# Green Synthesis of Gold and Silver Nanoparticles from Plant Extracts and Their Possible Applications as Antimicrobial Agents in the Agricultural Area

**DOI:** 10.3390/nano10091763

**Published:** 2020-09-07

**Authors:** Luis Castillo-Henríquez, Karla Alfaro-Aguilar, Jeisson Ugalde-Álvarez, Laura Vega-Fernández, Gabriela Montes de Oca-Vásquez, José Roberto Vega-Baudrit

**Affiliations:** 1National Laboratory of Nanotechnology (LANOTEC), National Center for High Technology (CeNAT), San José 1174-1200, Costa Rica; luis.castillohenriquez@ucr.ac.cr (L.C.-H.); jei.ugalde@gmail.com (J.U.-Á.); gmontesdeoca@cenat.ac.cr (G.M.d.O.-V.); 2Chemistry School, National University of Costa Rica, Heredia 86-3000, Costa Rica; karla.alfaro30@gmail.com (K.A.-A.); laly.vega@hotmail.com (L.V.-F.)

**Keywords:** agricultural industry, antibacterial, antimicrobial, green synthesis, gold, nanobiotechnology, nanoparticles, silver, sustainable development

## Abstract

Currently, metal nanoparticles have varied uses for different medical, pharmaceutical, and agricultural applications. Nanobiotechnology, combined with green chemistry, has great potential for the development of novel and necessary products that benefit human health, environment, and industries. Green chemistry has an important role due to its contribution to unconventional synthesis methods of gold and silver nanoparticles from plant extracts, which have exhibited antimicrobial potential, among other outstanding properties. Biodiversity-rich countries need to collect and convert knowledge from biological resources into processes, compounds, methods, and tools, which need to be achieved along with sustainable use and exploitation of biological diversity. Therefore, this paper describes the relevant reported green synthesis of gold and silver nanoparticles from plant extracts and their capacity as antimicrobial agents within the agricultural field for fighting against bacterial and fungal pathogens that can cause plant, waterborne, and foodborne diseases. Moreover, this work makes a brief review of nanoparticles’ contribution to water treatment and the development of “environmentally-friendly” nanofertilizers, nanopesticides, and nanoherbicides, as well as presenting the harmful effects of nanoparticles accumulation in plants and soils.

## 1. Introduction

Green chemistry has been developed as an alternative to the use of environmentally harmful processes and products due to the serious consequences that the world is facing and the limited available time to find effective solutions [1,2,3]. According to Menges, it is suggested that green chemistry could have saved USD 65.5 billion by the end of 2020 [4].

Chen et al. stated that circular economies (i.e., an economy that gradually decouples economic activity from finite resources consumption) should always aim to balance economic growth, resource sustainability, and environmental protection [5]. The challenge for biodiversity-rich countries and scientists working in these countries is to collect and convert knowledge from biological resources into processes, compounds, methods, and tools, which need to be achieved along with sustainable use and exploitation of biological diversity [6,7,8]. In addition to that, biodiversity exploration has been presented to the international scientific community as a promoter of the responsible use of nature, and as a means of obtaining non-harmful components as well. For this reason, different strategies have been sought to contribute to this field through the use of green processes, such as the creation of nanoparticles (NPs) from plant extracts [9,10,11].

NPs are a wide range of materials with dimensions below 100 nm, which can be used in various applications, such as medical, pharmaceutical, manufacturing and materials, environmental, electronics, energy collection, and mechanical industries, due to their multiple properties [12,13,14,15]. In general, NPs can be found as carbon nanotubes, quantum dots, nanorods, nanocapsules, nanoemulsions, fullerenes, metallic NPs, ceramic NPs, and polymer NPs [15,16]. Regarding the metallic NPs, their outstanding properties have caused the development of different methodologies for their synthesis, where gold (Au) and silver (Ag) NPs prepared from plant extracts are of great interest for the researchers in their attempt to develop suitable antimicrobial agents for agriculture [17,18,19,20]. Besides, these initiatives are considered as low-cost processes that allow avoiding toxic-generating products and benefit the agricultural activity. It is estimated that the preparation of one kg of silver nanoparticles (AgNPs) would cost about USD 4 million, while one kg of raw Ag costs around USD 14,000 [21,22].

In 2009, Raveendran et al. published one of the first green synthesis methods of metal NPs. They employed an aqueous starch solution subjected to heating, silver nitrate (AgNO_3_), and glucose as the green reducing agent [23]. After that, researchers like Iravani and Kumar et al. presented high-quality review papers regarding the synthesis of metallic NPs using plant extracts as a green chemistry approach [24,25]. Since then, the synthesis of metal NPs has been performed by different research groups based on a variety of plants and their structures. Logeswari et al. developed an eco-friendly synthesis of AgNPs from plant powders of *Solanum tricobatum*, *Syzygium cumini*, *Centella asiatica*, and *Citrus sinensis*, while Yang et al. biosynthesized gold nanoparticles (AuNPs) using an agricultural waste mango peel extract [26,27]. Verma et al. and Bagherzade et al. showed the antimicrobial activity of metal NP obtained through green synthesis using *Azadirachta indica* leaves and *Crocus sativus* L. extracts, respectively [28,29].

Due to the nanotechnological boom, unusual physical, chemical, and biological methods have been developed for the synthesis and production of metal NPs [30,31,32,33,34,35]. Therefore, the novelty of this paper lies in describing the relevant reported green synthesis of AuNPs and AgNPs from plant extracts and their capacity as antimicrobial agents within the agricultural field for fighting against bacterial and fungal pathogens that can cause plant, waterborne, and foodborne diseases. Moreover, this work makes a brief review of AuNPs and AgNPs’ contribution to water treatment and the development of “environmentally-friendly” nanofertilizers, nanopesticides, and nanoherbicides, as well as describing the harmful effects of NPs accumulation in plants and soils.

## 2. Methods for Obtaining Plant Extracts

Extraction methods are the first step for the separation of plant metabolites from the raw materials. In order to carry out an extraction process, some basic parameters need to be considered since these influence the quality of an extract [36]. The extraction of the interest components is largely dependent on the selected part from the plant material, as well as the solvents, which need to exhibit ease of evaporation, and the inability to chemically modify the solutes. Since the final product will retain traces from the employed solvent, the latter must have low toxicity. Additionally, selecting an extraction method involves addressing the length of extraction, temperature, solvent’s pH, solvent-to-sample ratio, and particle size of the raw materials. These factors can induce variations in the metabolite composition of the extracts [37].

The main extraction methods for the synthesis of AuNPs and AgNPs are (a) solvent-based extraction, (b) microwave-assisted extraction, and (c) maceration extraction [38]. However, ultrasound-assisted extraction is an important alternative to improve process efficiency. The ideal extraction method should be cost-effective, simple, less time-consuming, and carried out with ease in any laboratory [39].

### 2.1. Solvent-Based Extraction

This is the most widely used method for obtaining natural products. This method separates plant soluble metabolites using a suitable solvent and discards the insoluble materials [40]. Before starting, it is important to have information about the plant properties, such as the particle size of the raw materials and constituent components to select a proper solvent for the extraction. Besides, defining solvent-to-solid ratio, extraction temperature, and the time required for the process are also important. Factors that enhance the diffusion and solubilization of the solutes can increase the efficiency of the experiment [41]. The extraction process follows several stages, where initially the solvent penetration into the solid matrix is necessary to establish interactions with the available surface. Later, the solute must diffuse out of the solid matrix to collect it [42].

Recently, the application of green solvents (e.g., ethyl lactate and ionic liquids) has caught attention in different disciplines. These solvents are seen as non-toxic, biocompatible, and biodegradable alternatives to the conventional ones. In addition to that, they are easier to prepare and are cost-effective. Some advances regarding green solvent technologies are deep eutectic solvents (DESs), natural deep eutectic solvents (NDESs), ionic liquids (ILs), surfactants, and bio-derived solvents [43,44].

### 2.2. Microwave-Assisted Extraction

Microwave energy is employed for the partition of analytes from the sample into the solvent. This is caused by rapid heating due to the interaction with polar and polarizable compounds, such as water and the ones present in the plant matrix sample. In these, process heat is transferred by conduction, allowing materials to reach the necessary level of energy [38,45]. In addition to that, heat and mass transfer occur in the same direction, which results in a synergistic effect that accelerates and improves the recovery of analytes. Therefore, its implementation reduces the extraction time and solvent volume compared to other methods [46]. Aside from that, studies have shown other advantages, such as increasing the extract yield and selective heating of vegetal material [47].

Moreover, there are two variants for this method. Solvent-free extraction is usually employed for volatile compounds, while the solvent-based technique is recommended for non-volatile substances. Nevertheless, it is necessary to take into consideration two important aspects. In the first place, special concerns have to be foreseen for preventing samples’ thermal degradation, and second, this method is mostly limited to small-molecule phenolic compounds [48].

### 2.3. Maceration Extraction

This process can be achieved by following three basic steps: (a) grinding the plant in small pieces, (b) adding the appropriate solvent in a closed vessel, which will determine the type of compound that is going to be extracted from the sample, allowing soaking for three days at room temperature, and (c) press or strain by filtration to separate the liquid phase [49,50]. Although this may be considered the easiest and simplest method, the requirement of large volumes of organic solvents generates organic wastes that have become an issue, makingnecessarya proper chemical waste management process [51]. In addition to that, this conventional method for natural product extraction takes a long time, showing low extraction efficiency. The process intends to soften and break the raw material’s cell wall to propitiate the release of soluble phytochemicals. However, its use for extracting thermolabile components can be seen as an advantage [52].

### 2.4. Ultrasound-Assisted Extraction

Usually called ultrasonification, this method exhibits multiple advantages over other techniques, such as low energy consumption, less solvent required, lower temperatures and extraction times. This method is considered one of the easiest plant extraction techniques because it requires a smashed sample placed in an ultrasonic bath with a suitable solvent. It uses ultrasonic wave energy (20–2000 kHz) for extracting natural compounds from raw material, increasing the surface contact between the solvent and the sample [52]. Ultrasound enhances the solvent’s ability to penetrate the cells, which improves the efficiency of the process by disrupting the physical and chemical properties of the sample. Its influence on the solvent accelerates the dissolution and diffusion of the solute, facilitating the release of compounds. The method represents a cost-effective alternative for the scale-up of the industrial extraction of phytochemicals [53].

## 3. Green Synthesis Methods of AuNPs and AgNPs from Plant Extracts

Green chemistry is an emerging area that fosters the implementation of a set of principles intended to reduce the utilization and generation of chemical hazardous wastes [54]. As a result of that, green methods reduce the industrial labor impact on the ecosystem. Through their development, scientists are providing possible solutions to costly processes and hazardous materials encountered when using traditional physicochemical synthesis methods [55,56,57,58]. The employment of environment-friendly solvents and reagents, reducing high energy consumption methods, using non-toxic biomolecules, such as DNA, proteins, enzymes, carbohydrates, as well as plant extracts, allow synthesizing biocompatible metallic NPs by reducing metal ions in aqueous solutions [59,60].

AuNPs and AgNPs are considered noble metal NPs (i.e., metals resistant to corrosion and oxidation) and have caught the attention due to their unique physicochemical and biological properties [61,62]. These metal NPs have attractive attributes, such as high electrical and thermal conductivity, chemical stability, high catalytic activity, and the most relevant for agriculture—the antimicrobial activity against a wide diversity of microorganisms. These characteristics have a close relation to their nanosized particles, tunable shape, and surface morphology [61].

According to Jamklande et al., there are two kinds of synthesis methods for obtaining AuNPs and AgNPs, depending on the starting material for their preparation (Figure 1) [63]. In the first place, the top-bottom synthesis path is employed when raw material is at larger scales than the nano, forcing to break down its particles by grinding, milling, lithographic techniques, or thermal ablation. This particle size reduction techniques involve an enormous consumption of energy and can cause surface imperfections in NPs, resulting in significant effects on their physicochemical properties [58,63,64]. On the other hand, the bottom-up synthesis path, also known as the “self-assembly approach”, involves chemical and biological methods where atoms grow in nucleation centers to form NPs. Biosynthesis of AuNPs and AgNPs is a cost-effective type of bottom-up approach that allows an important amount of NPs’ formation in a short time. In this greener path, the obtained NPs possess minimum defects and present a homogenous chemical composition [63,65,66,67,68,69].

The use of plant extracts is increasing in usefulness and is conceived as an environmentally and economically friendly alternative for the synthesis of AuNPs and AgNPs by several techniques, such as using AgNO_3_ or auric chloride (HAuCl_4_) at room temperature within a few minutes to a couple of hours [65,71]. In addition to that, this synthesis method is faster compared to using bacteria or fungi [67]. Extracts can be obtained from multiple parts or products of the plant, such as leaves, bark, stem, shoots, seeds, latex, secondary metabolites, roots, twigs, peels, fruits, seedlings, essential oils, and tissues. They constitute a rich source of polyphenols, flavonoids, sugars, enzymes, and proteins. These phytochemicals are extracted and directly employed as reducing and stabilizing agents for the extracellular biosynthesis of metallic NPs, replacing potentially hazardous chemicals like sodium borohydride (NaBH_4_) [61]. However, the specific mechanism for this phenomenon has not yet been elucidated due to the great variety of phytoconstituents present in the extracts. Although polyphenols, organic acids, and proteins are considered as the main reducing agents, it is expected that the different phytochemicals work synergistically [61]. In general, this method can represent a cost-effective suitable option for large-scale production processes [72].

Moreover, these unconventional synthesis methods of AuNPs and AgNPs have the advantage of producing large quantities of NPs that are free from contamination and possess better-defined size and morphology. Regarding their morphology, fabrication through plant extract-mediated synthesis results in NPs with a more energetically favorable spherical shape, which provides the necessary reactivity for different applications, including agricultural activity [73,74]. Four studies have reported the synthesis of spherical AgNPs through different extracts like the ones from *Tribulus Terrestris* fruit, *Alternanthera dentate* leaves, *Acorus calamus* roots, and *Boerhaavia diffusa* species whole plant [75,76,77,78].

Other studies have produced NPs from various plant extracts, where, also, differences in shape are expected to happen when using different structures from the same plant for the extraction process [79,80]. However, as reported by Rajakumar et al., the use of *Eclipta prostrate* leaves for AuNPs synthesis produced triangle, pentagon, and hexagon shapes [81]. A disadvantage of these methods is that raw material’s nature limits the set of conditions under which they can be used, and this can impact NPs’ formation. Therefore, it is necessary to provide well-defined specifications regarding temperature, pH, metallic solution composition, and the reaction time as well [73].

Arreche et al. studied two commercial brands of yerba mate (*Ilex paraguariensis*) for the preparation of aqueous extracts to synthesize AgNPs at room temperature using AgNO_3_ (Figure 2). The obtained NPs were spherical, hexagonal, and triangular, with an average particle size of 50 nm and surface plasmon peak at 460 nm. Additionally, the antimicrobial activity was evaluated against *Escherichia coli* and *Staphylococcus aureus.* The minimum inhibitory concentrations required for *E. coli* were 7.66 and 17.66 μg·mL^−1^ using the treatment brand 1 and brand 2, respectively. On the other hand, the values for *S. aureus* were 23.25 and 50.60 μg·mL^−1^ for the treatment brand 1 and brand 2, respectively. The study suggests that polyphenols present in yerba mate leaf extract take action as a reducing agent and stabilizer of the NPs [82].

Besides, Sasidharan et al. used the pericarp of *Myristica fragans* fruit extract for the eco-friendly synthesis of AgNPs. In this approach, the aqueous fruit extract of the plant fulfilled reducing and stabilizing functions for the preparation, and the obtained AgNPs exhibited good catalytic and antibacterial activities [83]. Alkhalaf et al. conducted a study to identify the effect of the green synthesis of AgNPs from a *Nigella sativa* plant extract, resulting in NPs that exhibit antioxidant activity [84].

On the other hand, different attempts have successfully synthesized AuNPs, as well, through the application of a green process. Kesarla et al. worked on a green synthesis method using an aqueous extract of a fine powder of *Terminalia bellirica* dry fruit pericarp, which performed reducing and stabilizing functions. The experiment required to add one gram of the powder to 100 mL of deionized water, heat and maintain the temperature at 90 °C for 1 h, then cool and filter using a 0.2 mm cellulose nitrate membrane filter. After that, the freshly prepared extract was added to 2 mL of HAuCl_4_ 1 mM and mixed vigorously. The synthesis occurred almost instantaneously, taking less than 10 s to obtain the NPs, inferred from the immediate color change from yellow to reddish pink. The rapid reduction and stabilization can be explained by the high levels of polyphenols in *T. bellirica*. Additionally, AuNPs’ formation was confirmed by UV-Vis spectroscopy at 530 nm [85].

Some studies have synthesized and characterized both NPs. Sk et al. synthesized them using *Malva Verticillata* leaves’ aqueous extract. AuNPs were found to have outstanding catalytic activity toward the hydride transfer reduction of the aromatic nitro Schiff bases, while AgNPs displayed interesting antibacterial activity [86]. Nadagouda et al. reported a great presence of phenolic compounds in turmeric extract, cyanides, and polyphenols in blueberry, pomegranate, and blackberry, which are responsible for the antioxidant activity and the formation of AuNPs and AgNPs [87].

## 4. AuNPs and AgNPs Applications in Agriculture

In general, the synthesis of NPs is of great interest because of their unique properties that can be incorporated into composite fibers, biosensor materials, cryogenic super-conducting materials, cosmetic products, and electronic components [88]. However, due to climate change and the depletion of natural resources, the synthesis of AuNPs and AgNPs from plant extracts, and even more by agricultural wastes, is a major topic for encouraging sustainable development in agro-industrial labors. Since plants are the basis of this green synthesis, the created NPs can be used in many agroindustry-related processes, from the application in the soil to the food chain, due to their low toxicity [89,90,91].

Nanotechnological food and agricultural applications were proclaimed in June 2009 in a joint venture of the Food and Agricultural Organization (FAO) and World Health Organization (WHO), with the inclusion of wide-ranging fields, such as nanostructured ingredients, nanosized biofortification, food packaging, nanocoating, and nanofiltration [92]. NPs may also act as “magic bullets”, containing nutrients or other substances, such as beneficial genes, and organic compounds, which are targeted to specific plant areas or structures to enhance their productivity. Thus, NPs represent smart nano-delivery systems for agricultural administration, specifically on crop nutrition [93].

Regarding direct applications of AuNPs and AgNPs in agriculture, many pieces of researches on this field have been focused on seed germination, root elongation, and plant responses towards the presence of metal NPs, like cellular oxidative stress or cytotoxicity [89,94]. In addition to that, metal NPs can be employed for nano-fertilizers and nano-pesticides development [95]. Indirect applications based on the antimicrobial activity of the NPs are mostly related to food packaging [93]. The mentioned applications have been widely addressed in the agro-industries in a great variety of products containing NPs of these metals with a particle size that ranges from 100–250 nm, making them more soluble in water, and increasing their activity [96].

### 4.1. Antimicrobial Properties

Metal NPs from green synthesis can be used as antioxidants, biosensors, and for heavy metal detection as well [97,98,99]. In general terms, their unique physicochemical properties, such as their ability to bind biomolecules, large surface area to volume ratio, high surface reactivity, easy to synthesize and characterize, reduced cytotoxicity, and their capacity of enhancing gene expression for redox processes, allow using as antimicrobial agents against plant disease pathogens and others that can cause foodborne diseases [100,101,102,103]. A great variety of plant extracts used for generating AuNPs and AgNPs have been processed and reported their potential antimicrobial activity against bacterial and fungal plant pathogens (Table 1) [104,105].

However, under conventional synthesis methods, these NPs exhibit nearly no antimicrobial activity [117]. The difference may lie in a synergistic effect due to the combination of AuNPs or AgNPs and the plant extracts, which provide high concentrations of steroids, sapogenins, carbohydrates, and flavonoids that act as reducing agents of ions and cover agents, contributing to NPs high stability [118]. Antimicrobial application is primarily due to their ultra-small size and shape (250 times smaller than bacteria), enabling an electrostatic interaction between the Au or Ag from the NPs, and the negative charge on the cell wall or surface of microorganisms, resulting in a distortion of membrane’s power functions, such as permeability, osmoregulation, electron transport, and respiration, leading them towards cellular death. The previous depends also on surface availability; thus, smaller NPs will present greater bactericidal effect by binding with the large surface area of the bacteria’s cell membrane [119,120,121].

Nevertheless, the exact antimicrobial mechanism for AuNPs and AgNPs is still poorly understood, for which several theories have been proposed. In the first place, NPs have a strong preference for reacting with sulfhydryl and phosphorus groups on the cell wall, causing great damage that results in the release of bacterial cell components [122]. Another hypothesis states that NPs can penetrate the bacterial cell membrane and attach to NADH dehydrogenases, generating a high amount of reactive oxygen species (ROS), which cause the depletion of ATP and interrupts the respiratory chain. These radicals have the capacity to interact with proteins, sulfur, phosphorus-containing cell constituents, and DNA, destroying them (Figure 3) [122,123]. Finally, NPs’ release of Au or Ag ions may contribute to their antimicrobial activity since DNA loses its replication activity, and proteins get inactive after interacting with these ions. This biocidal effect is considered to be size and dose-dependent; a higher concentration of NPs have demonstrated to interact more with cytoplasmatic organelles and bacterial nucleic acids [124,125].

Moreover, the antimicrobial activity shows differences related to bacterial cell wall composition. AuNPs and AgNPs exhibit higher activity against Gram-negative bacteria than to Gram-positive. The latter possesses a thick layer of peptidoglycan, consisting of linear polysaccharide chains cross-linked by short peptides, which creates a hard rigid structure for NPs to penetrate, while Gram-negative has a thinner that represents a feasible composition to pierce [127,128]. Nevertheless, works from Grace et al. and Padalia et al. reported that NPs’ antimicrobial activity could be enhanced by surface modification through coating with aminoglycoside antibiotics, increasing their activity range to Gram-positive bacteria as well [129,130]. Wang et al. discovered that gentamicin enhanced bactericidal toxicity of AgNPs against foodborne bacteria *S. Aureus*, *E. coli*, and gentamicin-resistant *E. coli* because the drug promotes NPs dissolution, increasing Ag ions concentration, and promoting bacterial growth inhibition and death [131]. It is worth to mention that this can be a possibility for agricultural labors that employ antibiotics. Although different research groups are raising awareness of the problem and encouraging to dismiss the use of antibiotics in any other field different than medical, under a controlled environment, and highly regulated application, this method can be highly effective and reduce drug resistance issues caused by this human practice [132].

Many research papers in which AuNPs have shown promising antimicrobial activities have highlighted a majority spherical shape character of the NPs. Nevertheless, rod-shaped, triangular, hexagonal, and cubic NPs have also been found as part of the obtained mixture [133]. Thangamani et al. synthesized AuNPs using *Simarouba glauca* leaf extract. NPs’ size and shape were sensitive to leaf broth concentration; particles tended to decrease in size with an increase in leaf broth concentration, while different morphologies were obtained, such as a mixture of the prism and spherical-like particles. Aside from that, they assessed the antimicrobial activity by testing them against Gram-positive and Gram-negative organisms. The antimicrobial assay showed better results for *S. aureus*, *S. mutans*, *B. subtilis*, *E. coli*, *Proteus vulgaris*, and *K.a pneumonia* [134].

*Euphorbia hirta* aqueous leaf extract was used for biosynthesizing AuNPs by Annamalai et al. NPs sizes ranged from 6–71 nm, which greatly contributed to their highly effective antimicrobial activity against bacterial strains of *E. coli*, *P. aeruginosa*, and *K. pneumonia*. The study evaluated concentrations of 1.25–200 ug/mL, where 200 µg/mL completely inhibited the bacterial growth [135]. Additionally, Folorunso et al. investigated the antimicrobial potency of synthesized AuNPs from the leaf extracts of *Annona muricata*. The resulted NPs showed a spherical structure with an average size of 25.5 nm and high antibacterial activity against *Clostridium sporogenes*, where the efficacy increased as the NPs’ concentration increased [136].

In another approach, Montes de Oca et al. evaluated the impact of AgNPs’ usual concentrations in nature soils grown with Arabian Coffee in customary and organic operating systems. In this study, biomass, extracellular enzyme activities, and the diversity of the soil microbial community were assessed by a microcosm experiment as a function of time. After 7 days of incubation, the increase in the microbial biomass was independent of NPs concentration [137]. In contrast, after 60 days, there was a decrease in Gram-positive and actinobacterial biomass in soils at all the evaluated AgNPs concentrations. The physicochemical properties of the soil and the enzymatic activities were not affected by AgNPs. Within the composition of the microbial community, only a few differences were observed in abundance relative to the phylum level and gender in the fungal community [137]. The results indicated that the environmental factors of AgNPs affected microbial biomass but had a low impact on microbial diversity and might have a poor effect on soil biogeochemical cycles by extracellular enzyme activities [137,138].

A study by Lediga et al. functionalized AgNPs with the extracts of *Sclerocarya birrea* and *Eucomis autumnalis*. Remarkable antimicrobial properties against two Gram-negative and two Gram-positive bacteria were observed. Both the *S. birrea* and *E. autumnalis* AgNPs exhibited negligible or low toxicity to plants [139]. Additionally, Shahryari et al. formed AgNPs with an average size of 35 nm using the sumac aqueous extract through a green synthesis method for controlling bacterial canker disease of stone fruit trees caused by *Pseudomonas syringae*. The results revealed that different NPs concentrations reduced the severity of the disease, reaching the highest decrease at 100 ppm NPs concentration [140].

Moreover, the antimicrobial activity can be used to fight and control fungal pathogens. Jebril et al. green synthesized AgNPs using the leaf extract of *Melia azedarach* to evaluate its antifungal activity. The researchers reported high efficiency against *Verticillium dahliae* in eggplant at a concentration of 20 ppm [141]. Besides, Balasubramanian et al. produced AuNPs from the leaf extract of *Jasminum auriculatum*, which displayed antifungal activity against *Aspergillus fumigatus*, a very common pathogen in cultivated fields [142].

Furthermore, nanocoatings containing AuNPs and AgNPs have been applied to food contact surfaces and for packaging, taking advantage of their antimicrobial properties. This technology slows down chemical oxidation and degradation processes induced by microorganisms, extending the shelf life of packed crops and other food products [143,144].

### 4.2. Water Treatment

Water quality is fundamental for developing agricultural labors, but also wastewater treatment is worth bringing into the discussion since agricultural labors use about 70% of the renewable water sources worldwide [145]. Heavy metals, fertilizers, and pesticides employed for agricultural activity have reduced the supply of pure water for drinking and crop irrigation [146]. Nanotechnology can take an important part of a sustainable productivity system, ensuring its quality and purity. The most attractive and cost-effective nanomaterials for environmental protection and water remediation processes are derived from noble metals. In this application, the high surface area to volume ratio, chemical stability, and enhanced catalytic activity properties from plant extract-mediated green synthesized AuNPs and AgNPs can be employed for water monitoring, purification, drinking water treatment, and agriculture wastewater treatment [147,148].

Moreover, research groups have taken advantage of the different properties exhibited by these NPs, including their high reactivity for identifying toxic substances, such as pesticides and heavy metals (e.g., lead, mercury and cadmium), by incorporating them into sensors for the rapid detection of these chemicals [149,150]. Water and wastewater detoxification can be achieved by adsorption, photocatalytic degradation, and nanofiltration techniques using NPs as well [151]. Different authors have described the process of pesticide mineralization in water using AuNPs and AgNPs, such as chlorpyrifos, malathion, and atrazine [152]. Pesticides extraction is achieved by their adsorption onto NPs, which retain them on their surface, interacting for long periods until the complex precipitates. Therefore, these NPs represent a suitable, convenient, and cost-effect means of removing pesticides for either drinking water or irrigation labors [152,153]. Moreover, AuNPs and AgNPs are considered an interesting approach for heavy metals elimination in water due to their high adsorption capacity [154].

In addition to that, water pollution with bacterial pathogens represents a high risk for water-borne, food-borne, and plant diseases. The antimicrobial properties of metal NPs have been reported to be effective in this type of water purification [155]. Francis et al. synthesized AuNPs and AgNPs from the leaf extract of *Mussaenda glabrata* and evaluated their capacity to inhibit pathogenic microorganisms. The NPs showed outstanding antimicrobial activity against *P. aeruginosa*, *E. coli*, *A. niger*, and *Penicillium chrysogenum* [156]. Besides, their catalytic capacities make them suitable for dye degradation, such as reported by Veisi et al. in their research, where green synthesized AgNPs from the leaf extract of *Thymbra spicata* decreased different dyes, such as nitrophenol, rhodamin, and methylene blue [157].

### 4.3. Development and Delivery of Nanofertilizers, Nanopesticides, and Nanoherbicides

Agricultural labors are unimaginable without the use of fertilizers for crop nutrition, as well as pesticides and herbicides for protecting crops and plants from several insect pests and plant diseases that continuously hamper the activity, causing high economic loss and food security issues. However, the use of these types of substances has resulted in huge ecotoxicological effects, microbial resistance to agrochemicals, and even accumulation in the human organism [158]. In this sense, nanotechnology can provide specific applications for sustainable development through nanofertilizers, nanopesticides, and nanoherbicides by developing smart delivery systems [159]. Scientists have designed nanocapsules using AuNPs and AgNPs to provide better control of nutrients, increasing the efficacy compared to conventional formulations for boosting crop production. This implies a potential reduction of about 10–15 times in the applied dose and frequency. Nanocapsules exhibit a high-loading capacity with a gradual release of the substances, representing beneficial aspects for plant growth since the presence of these NPs in soil may affect root nutrient uptake and water transport, as has been addressed in the following sections [160,161,162].

AuNPs and AgNPs-based nanofertilizers have been developed to synchronize nutrient release with plant uptake. This system reduces nutrient loss, soil and groundwater contamination, and chemical reactions with water, soil, and microorganisms that transform them into unuseful or toxic substances for plants, helping to maintain the soil’s fertility [163,164,165]. Kang et al. applied 5 mg/L AgNPs fertilizer suspension to red ginseng shoot three times per day at 14-day intervals. After harvesting, they reported that the nanofertilizer had enhanced the ginsenoside content [163].

Nanoencapsulated pesticides and herbicides show enhanced properties in terms of solubility, specificity, permeability, and stability because the nanostructure protects the active substance from early degradation and provides pest control for longer periods [166]. Moreover, control of plant disease-causing phytopathogens, such as bacteria and fungi, can also be achieved by spraying a NPs solution directly on grains, seed, or foliage to inhibit the invasion of plant pathogens [167]. AgNPs green synthesis from *Fusarium solani* was done by El-Aziz et al. to evaluate their impact on grain borne fungi. The outcome of this research was that sprayed NPs solutions of 4% caused a 0% frequency of fungal pathogens [168]. Gnanadesigan et al. biosynthesized AgNPs using the leaf extract of *Rhizophora mucronata* to evaluate its potential larvicidal activity against *Aedes aegypti* and *Culex quinquefasciatus*, two vectors that affect workers in agricultural fields, causing dengue and filariasis [169].

The adoption of high-tech agricultural systems can reduce or even eliminate the negative environmental influence of modern agriculture, as well as enhancing the quality and quantity of crop production [170]. Although NPs for these applications provide a lot of benefits, their different properties give them different toxicities, which need further research [171,172,173].

## 5. Interactions of NPs with Plants: Accumulation and Harmful Effects of NPs in Plants and Crops

AuNPs and AgNPs are used in different commercial products for agricultural labor, such as nanofertilizers, nanopesticides, and nanoherbicides. The nanosize to which many useful properties are attributable constitutes the factor that can enhance their adverse effects too [174]. Recently, questioning is being raised about their toxicity regarding short- and long-term environmental exposure. Special attention should be paid to the interaction between NPs and plants (e.g., crops) when these materials are used within the agricultural field [175].

Ecotoxicological research is required to demonstrate the environmental consequences of plants constantly exposed to these nanomaterials. NPs in contact with plants can enter to the cellular system, translocate their shoots, and accumulate in different parts. In addition to that, there is a potential risk of cycling through different trophic levels [176]. AuNPs and AgNPs accumulation in plants have shown various effects on the transpiration and respiration rate, which alters the photosynthesis process. At the cellular level, these NPs can alter plants’ growth rate and chlorophyll levels, while on the sub-cellular level, investigations have reported organelles modifications and NPs deposition close to the plasmatic membrane [177]. Moreover, the uptake, translocation, and accumulation depend on NPs’ size, type, chemical composition, and stability, as well as the plant species [178].

Judy et al. used the model organisms *Nicotiana tabacum*, *Xanthi*, and *Manduca sexta* to evaluate plant uptake and possible trophic transfer upon the exposure to AuNPs of 5, 10, and 15 nm. Their results confirmed trophic transfer and biomagnification of the NPs from a primary producer to a primary consumer by different mean factors related to the NPs’ size treatments [179]. Hashimoto et al. found that accumulated AgNPs could translocate to roots and shoots of two terrestrial agro-crops—*Vigna unguiculata* and *Triticum aestivum*. Recently, it has been demonstrated that AgNPs under aerobic soil conditions are able to maintain their intact nature (88%), while a transformation to Ag_2_S can also occur in the same extension [180].

While it is not clear how metal NPs affect the environment, some studies reveal that plants overexposed to them may reveal pathways involved in the cytotoxicity. Proteomic studies on *Oryza sativa* (Asian rice) with AuNPs and AgNPs have increased protein precursors for oxidative stress tolerance, calcium regulation and signaling, apoptosis, and other kinds of damages [181]. Additionally, Vanini et al. developed research to study the proteomic profile of *Eruca sativa* exposed to AgNPs and bulk Ag. Seedlings were treated for five days with different concentrations of AgNPs and AgNO_3_, resulting in changes in proteins involved in redox regulation and the sulfur metabolism for both cases. However, further analysis revealed an altered number of proteins in the endoplasmic reticulum and vacuole of plant cells caused by AgNPs [182]. Kaveh et al. studied the model agro-crop *Arabidopsis thaliana* and reported the phytoaccumulation of AgNPs [183]. Another approach developed by Taylor et al. described *Medicago Sativa* L. (alfalfa) tendency to accumulate metal NPs of different sizes [184].

A research conducted by Raliya et al. studied uptake, translocation, and accumulation of different AgNPs with sizes ranging from 30–80 nm delivered by aerosol application to a watermelon plant. The findings indicate that NPs could be taken up by direct penetration and transport through the stomatal opening. Besides, they observed translocation of NPs from leaf to shoots, which suggests they travel by the phloem transport mechanism (i.e., the mechanism of long-distance transport through plant’s sieve tube) [185]. It has been reported that Au is taken up in *A. thaliana* predominantly in an ionic form, having a significant role in seed germination and antioxidant system. However, other studies suggest that AuNPs’ exposure results in the upregulation of plant genes, causing downregulation of specific-metal transporters to reduce Au uptake [184]. This can be used for studying the limits of NPs in the environment.

Moreover, Qian et al. discovered that high concentrations of Ag could be overwhelming to *A. thaliana* seeds, which should not be exposed to AgNPs during its germination [186]. In contrast, Stampoulis et al. reported no toxic effect on seed germination and root elongation of *Cucurbita pepo* (zucchini) when exposing to AuNPs and AgNPs. However, in a 15-day hydroponic trial, the biomass and transpiration of the plants exposed to AgNPs were reduced by 75% and 41%, respectively, as compared to control plants and the corresponding bulk Ag powder. Additionally, zucchini shoots exposed to these NPs contained, on average, 4.7 greater Ag concentration than the ones from bulk solutions [187]. Germination studies in *Lolium perenne*, *Hordeum vulgare*, and *Linum usitatissimum* have shown to be affected at low concentrations of AgNPs but never fully inhibited [188]. This suggests that different mechanisms of action might occur across plant species, concerning the effect on germination [189].

Courtois et al. published an important study of the impact of Ag species introduced into the soil via sewage sludge. As mentioned before, AgNPs are incorporated into many conventional and novel products due to their special physicochemical and antimicrobial properties. However, the discharge of these products into wastewater causes an accumulation of AgNPs and Ag_2_S in sewage sludge. The major concern is related to land application of sewage sludge for agricultural purposes since soils receive a great source of contamination for plants and crops. Soil exposure to metal NPs may lead to changes in microbial biomass that can affect plant growth, causing physiological, biochemical, and molecular effects on them. Nonetheless, much is still unknown about the ecotoxicology of silver species, where several doubts are focused on the possibility of transfer along the trophic chain via accumulation in plants, and for that, research to evaluate the long-term impact of AgNPs on plants is ongoing [190].

## 6. AuNPs and AgNPs in Soils

Biological indicators are important parameters for evaluating soil quality since soil microbiota directly participates in different processes of this ecosystem, such as decomposition of inorganic matter and nutrient cycling. Therefore, any factor that alters soil microbial biomass will have an impact on soil sustainability [191]. Needless to say, the growing use of AuNPs and AgNPs due to their recognized antimicrobial activity has led to their accumulation in soil ecosystems, affecting its quality [192,193]. Although their environmental impact on the soil microbial community is still under consideration, several authors have concluded that the toxic effects on microbial communities are highly dependent on their concentration in the soil [194,195,196,197,198]. However, most studies have evaluated NPs at higher levels than actually occur in nature [199,200,201].

Dinesh et al. stated that NPs might have an impact on the soil in different ways. Firstly, a direct effect could be attributed to their properties, such as their antimicrobial activity, which might reduce soil microbiota mostly by generating ROS. In the second place, NPs could induce changes in toxins and nutrients bioavailability. Finally, the indirect effects might result from their interaction with natural organic toxic compounds, thus enhancing their toxicity [191]. In addition to that, different studies have proven that AgNPs affect microorganisms that promote plant growth and nutrient cycling in soils, such as *rhizobacteria*, *Pseudomonas fluorescens*, *Pseudomonas putida*, and others. A major concern is related to the inhibition of denitrifying bacteria, where studies have reported that at concentrations of 100 mg of AgNPs/kg of soil, can be observed a complete reduction of these colonies without recovery signals and with the consequence of the reduced conversion of nitrates to nitrogen [191,202].

NPs can affect other soil organisms, such as earthworms. Unrine et al. determined AuNPs bioavailability from soil to *Eisenia fetida*. They reported that NPs sizes ranging from 20–55 nm did not influence distribution among tissues in contrast to the ones smaller than 20 nm, which were more available. In addition to that, AuNPs could cause adverse effects on earthworm reproduction [203]. Bourdineaud et al. also used *E. fetida* to evaluate the transfer of AuNPs and AgNPs from soil to the earthworm tissues. In this study, the invertebrate was exposed to soil containing 2, 10, and 50 mg of NPs/kg of soil for 10 days. Both NPs showed similar transfer coefficients and induced the onset of oxidative stress that caused DNA modifications even at the lowest evaluated concentration [204]. Another study by Schlich et al. evaluated the effects of AgNPs on *Eisenia Andrei* earthworm reproduction. Ag uptake from the NPs was slightly higher compared to AgNO_3_, where both substances showed similar toxicities in the reproduction test. However, Ag uptake was not reported to inhibit reproduction [205]. In contrast, Ploeg et al. described the reduction in the reproduction of *Lumbricus rubellus* earthworms influenced by AgNPs, which also caused mortality upon long-term exposure [206].

In addition to NPs concentration in soil, ecotoxicological studies should take into consideration dissolution rate, size, surface area, electric charge, and their surface chemistry since these control NPs’ stability and their transport, giving relevant and more accurate outcomes about their toxicity. Furthermore, dispersibility is also considered as a key factor; agglomerates of NPs have shown less toxicity compared to well-dispersed forms. A way of promoting that is by using biodegradable polymers as stabilizing agents obtained through green methods, such as chitosan, wood cellulose, gelatin, among others [207]. Moreover, the behavior of NPs within soil systems is influenced by the presence of surface charged components, such as clay, which alters their association with the solid phase. This interaction between NPs and solid surfaces controls their transport along with the soil, which is influenced by environmental conditions and the NPs’ physicochemical properties. Particle retention in pore soils is especially relevant to the bigger NPs, whereas smaller particles tend to move more freely and can penetrate to reach groundwater [208].

Meier et al. presented the concern that anthropogenic activities could disrupt soil ecosystems, resulting in the reduction of its microbial health. In order to evaluate the previous, they exposed freshly collected sandy loam soil to solutions ranging from 0–2000 mg/kg of AgNPs. After that, they expanded traditional soil microbial analysis with genomics-based tests through the measure of alterations in community taxonomic structure and function using 16S-rDNA profiling and metatranscriptomics. The research group found that AgNPs affected bacterial taxonomic structure, as well as genes involved in heavy metal resistance. Besides, their presence induced some toxicity response pathways to become highly upregulated [209].

In Canada, AgNPs are employed for direct application in soils to reduce the utilization and degradation of conventional pesticides, provide micronutrients, increase crop yields, and control plant pathogens. Regarding that, Asadishad et al. evaluated the effects of AgNPs on the soil microbiota and enzyme activity of agricultural soils at different concentrations (1, 10, or 100 mg of AgNPs/kg of soil) over 30 days. At the end of that period, AgNPs inhibited selected enzymes at the concentration of 100 mg/kg, causing interesting changes in soil microbial community [210]. Another study by Li et al. described the impact of AgNPs on the soil. Ag_2_S is more likely to be the form in which silver is retained in soils. They examined Ag_2_S retention from 11 natural different soils and discovered that more than 99% of the NPs were retained irrespective of the soil properties. Since the retention of Ag_2_S in soils is conceived as a critical factor for their toxicity and availability to sustain life (e.g., plants), the results obtained by this group can be a good approach for explaining the differences in phytoavailability exhibited by soils compared to what is established in the literature for liquid media [211].

## 7. Conclusions

Green chemistry is an innovative and growing resource in the search for more environmentally friendly processes. Using plant extracts for the synthesis of metal NPs is a recently growing area of interest due to its benefit in comparison to the traditional physicochemical methods. AuNPs and AgNPs generated by green synthesis have potential applications in agriculture and agroindustry, especially as antimicrobial agents of certain microorganisms for which their efficacy has been scientifically proven. Although recent studies suggest that environmental concentrations of AuNPs and AgNPs affect microbial biomass with low impact on their diversity, further research needs to be addressed in order to determine the effects they could produce to the soil, plants, and the environment, in general, due to long-term exposure. Therefore, local and national regulatory institutions must establish guidelines and monitoring methods for better use of these nanotechnological advances.

## Figures and Tables

**Figure 1 nanomaterials-10-01763-f001:**
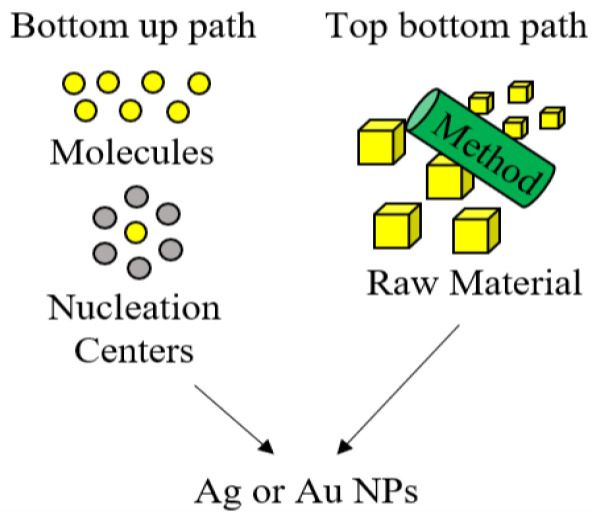
Synthesis of metal nanoparticles (NPs) from top-bottom and bottom-up paths [70].

**Figure 2 nanomaterials-10-01763-f002:**
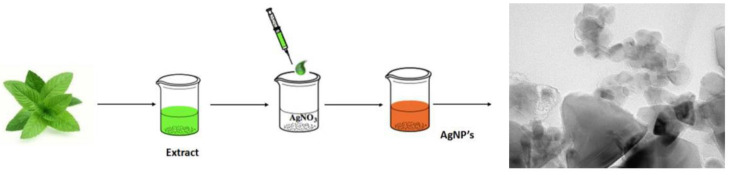
Synthesis of silver nanoparticles (AgNPs) using natural extracts from wastes.

**Figure 3 nanomaterials-10-01763-f003:**
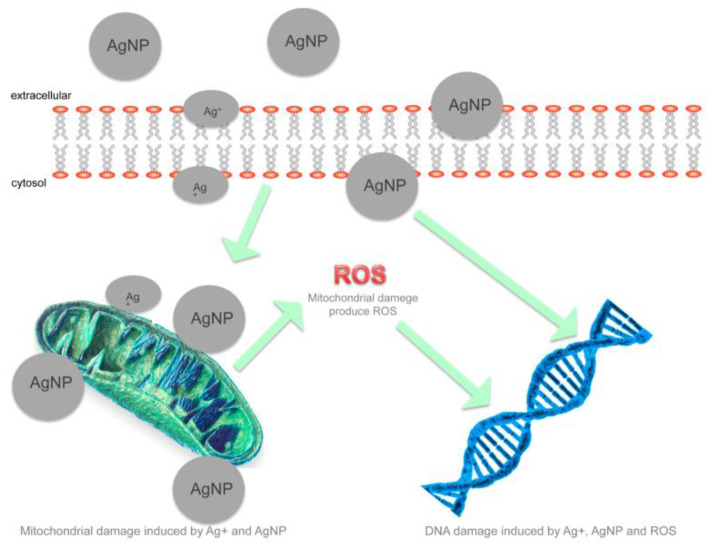
AgNPs antimicrobial mechanism through bacterial cell membrane penetration, interruption of the respiratory chain, and DNA damage by reactive oxygen species (ROS) [126].

**Table 1 nanomaterials-10-01763-t001:** Plant extract-mediated green synthesis of AuNPs and AgNPs with antimicrobial activity.

Plant Extract	Synthesized NPs	Target Pathogen	Ref
*Citrus limetta* peel	AgNPs	*Micrococcus luteus, Streptococcus mutans, Staphylococcus epidermidis, S. aureus, E. coli, Candida* spp.	[106]
*Luffa acutangula* leaf	AgNPs	*E. coli, Saccharomyces cerevisiae*	[107]
*Parkia speciosa* leaf	AgNPs	*E. coli, S. aureus, Pseudomonas aeruginosa, Bacillus subtilis.*	[108]
*A. indica* leaf	AgNPs	*S. aureus, E. coli.*	[109]
*Gomphrena globosa* leaf	AgNPs	*S. aureus, B. subtilis, M. luteus, E. coli, P. aeruginosa, Klebsiella pneumoniae.*	[110]
*Pedalium murex* leaf	AgNPs	*E. coli, K. pneumonia, Micrococcus flavus, P. aeruginosa, B. subtilis, Bacillus pumilus, S. aureus.*	[111]
*Musa acuminate* peel	AgNPs	*B. subtilis, S. aureus, P. aeruginosa, E. coli.*	[112]
*Caulerpa racemosa*	AuNPs	*Aeromonas veronii, Streptococcus agalactiae*	[113]
*Eclipta alba*	AuNPs	*E. coli, P. aeruginosa, B. subtilis, S. aureus.*	[114]
*Nepenthes khasiana* leaf	AuNPs	*E. coli, Bacillus* spp., *Aspergillus niger, Candida albicans.*	[115]
*Abelmoschus esculentus* pulp	AuNPs	*B. subtilis, Bacillus cereus, E. coli, M. luteus, P. aeruginosa.*	[116]

*A. indica: Azadirachta indica; M. luteus: Micrococcus luteus; S. aureus: Staphylococcus aureus; E. coli: Escherichia coli; P. aeruginosa: Pseudomonas aeruginosa; B. subtilis: Bacillus subtilis; K. pneumonia: Klebsiella pneumonia*.

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
