# Peer review of "Green Synthesis of Gold and Silver Nanoparticles from Plant Extracts and Their Possible Applications as Antimicrobial Agents in the Agricultural Area"

_nanomaterials, 2020, doi:10.3390/nano10091763_

Round 1
Reviewer 1 Report
The manuscript relates the advancements in the utilization of plant extracts as a step in the formation of metallic NPs for differing purposes, and the title of the paper draws focus on agricultural applications. While there is some application discussed in various agricultural systems, the applications component of the manuscript is not equally developed, and not well-balanced with the component of the manuscript discussing the development of NPs and the integration of plant extracts into NPs formation. The reviewer suggests either the authors provide more discussion on the differing applications in agri-food industries, with clear and focused listing of strengths and weaknesses versus other systems, or that they reduce the length of discussion on conventional methods of NPs formation so that the point of green production is made more significant/amplified.
Additionally, the authors' choice of wording throughout the manuscript at times is unclear in its intended meaning, and at some points is sensational, but not objectively clear. This limits reviewer's understanding of the authors' intended meaning/point, and draws away from the objective review of cited data/research.
The authors do provide a broad, extensive citations list, and figures tend to be very useful in graphically illustrating key points. The authors provide a global focus to the review in citations and inclusion of recommendations on NPs by FAO/WHO, amongst other positive attributes. The manuscript, finally, will be greatly improved by authors' cooperation with journal editorial staff in correcting various minor grammatical and punctuation errors, in addition to revision of wording at various points in order to improve manuscript clarity. See below for specific comments.
L15: benefit human activities, … What is the authors’ meaning of this phrase? Industries? Human health protection? The wording is unclear and should be revised.
L15: , and encourages the…
L29: Change to: Green chemistry has… (Remove Currently from beginning of sentence).
L34: The relationship of circular economies to the authors’ point is not clear/understood. The reviewer is somewhat familiar with the concept of circular economy but how does it relate here?
L36: countries, and scientists working in these countries,...
L45: Suggest revising to applications from areas.
L47: This is not an exhaustive list of NP types; authors fail to note emulsions on nanoscale can be engineered to form NPs, as a number of other types of NPs.Authors should either attempt a more comprehensive listing of classes, or focus on the metallics within the various classes, given the subsequent sentences.
L86: milli-continuous? Mis-spelling.
L93: recalls the importance… What does this mean? The authors’ wording is again unclear to the reviewer, and may be also to readers.
L96: While the authors’ meaning is not lost on the reviewer, in ancient times NPs were not being formed, and this wording should be discarded and replaced with phrasing indicating Ag-containing dyes, stains, fluids, etc.
L139-43: Suggest moving these sentences to the previous paragraph, just after the sentence ending in L135.
L145: What is the authors’ meaning of social benefit in this sentence? Energy consumption? Small business development? This wording is increasingly appearing to the reviewer to be ineffective and not appropriate when describing the reviewed processes
L154: … than some obtained…
L155: bioassays? What? The authors were discussing advantages of green production of metallic NPs, and this indicates a shift in focus with no transition or conclusion on the previous point. What bioassays? Do authors mean only the negatives of the green production of NPs? If so, this must be revised. Any way, the wording must be revised to clarify the authors’ point.
L194-5: Give examples of solvents; some alcohols are very useful as solvents, as is even steam, and these are likely even more biocompatible (steam, particularly, or ethanol) than any of the listed solvents.
L202: with both… (remove comma)
L204: What aforementioned? This is vague and not useful description. Replace with more clear, direct description of conditions that can demonstrate microwave is green.
Section 5.3: How is maceration distinct from solvent extraction? By authors’ description, it differs only in the potential need for filtration to remove particulate matter of macerated plants, but some solvent extraction procedures also rely to some degree on physical breaking of plant tissues to increase surface area to volume ratios in reaction vessels to improve total extraction. So how does one method really differ from the other? Current text does not really distinguish them.
L218: strongly arising? Meaning? Revise to, “... is increasing in usefulness…”
L227: derivative products… utility (delete aptitude)
Figure 3 conveys little useful understanding given its simplicity. Reviewer recommends its deletion or its significant revision to actually transfer more knowledge to readers on possible methods of formulation, sources of plant material, etc.
L304: is well illustrated by…
L370: bacteria.
L371: toxicity to what? Human cell lines? Animals? Indicate more clearly.
L411: and derivatives...
Author Response
Reviewer 1: Dear reviewer, thank you for your observations. The manuscript was reviewed for reducing wording. Additionally, I would like to mention you that we deleted several sections from the original manuscript as suggested by other reviewers. Instead, you’ll now find the following structure:
- Introduction: We clearly stated the novelty of our manuscript.
- Methods for obtaining plant extracts: we added more information to this section, including ultrasound-assisted extraction.
- Green synthesis methods of AuNPs and AgNPs from plant extracts: we added more references to this section which is almost 3 pages long.
- AuNPs and AgNPs applications in agriculture: we agree this section lacked of the need amount of references. Therefore, this section is now almost 5 pages long and contains about 84 references from the literature. Here we talk about the potential antimicrobial application of these NPs, and how that activity can be used for water treatment and pest control. In addition to that, this section briefly reviews how these NPs can work as delivery systems for nanofertilizers, nanopesticides, and nanoherbicides.
- NPs interaction with plants: here we develop harmfull effects caused by AuNPs and AgNPs due to their accumulation in plants’ tissues, as well as other issues such as NPs uptake, translocation, and trophic transfer.
- AuNPs and AgNPs in soil: similarly to the previous section, here we describe the mechanism of interaction between the NPs and soil’s surface, NPs alteration of soil microbial biomass and its impact on soil quality and sustainability. Additionally, we mention some research regarding harmful effects to other organisms such as earthworms.
- Modifications according to comments 1 and 2 can be seen in L15-16.
comment 3: L32.
comment 4: L37-L38.
comment 5: L40.
comment 6: L49.
comment 7: L51-L52
comment 8: deleted from manuscript.
comment 9: deleted from manuscript.
comment 10: deleted from manuscript.
comment 11: deleted from manuscript.
comment 12: deleted from manuscript.
comment 13: L188-190.
comment 14: L198.
comment 15: L106
comment 16: deleted from manuscript.
comment 17: deleted from manuscript.
comment 18: L97-111 and L127-136. Differenciation has been presented in terms of process eficiency, extraction time and type of solvent employed.
comment 19: L172
comment 20: L184-188.
comment 21: figure 3 was deleted as requested.
comment 22: L191-194.
comment 23: L352.
comment 24: L353-354.
comment 25: L497.

Reviewer 2 Report
The review developed by Castillo-Henríquez and colleagues provides an original scientific approach concerning the role of green synthesis (based on plant extracts) in the development of antimicrobial silver and gold nanoparticles and their use in agricultural industry. The work is based on a broad spectrum of current scientific literature with regard to the role of nanoparticles in different areas of human life, as well as methods of their synthesis using conventional and unconventional techniques. A little scientifically weaker is Chapter 5, where other effective methods for obtaining plant extracts, e.g. ultrasound-assisted extraction, should be considered. The authors' critical approach, to both the benefits of using silver and gold nanoparticles in agricultures and the possible risks associated with the phenomenon of NPs accumulation in crops and soils and direct cytotoxic effects on plant cells, deserves praise. It is also worth noting that the work is written in very good English. My general opinion, as a reviewer, is positive, so I recommend accepting the manuscript for publication in the Nanomaterials.
Author Response
Reviewer 2: Dear reviewer, thank you for your observations. The manuscript was reviewed and restructured taking into consideration your comments. Additionally, I would like to mention you that we deleted several sections from the original manuscript as suggested by other reviewers. Instead, you’ll now find the following structure:
- Introduction: We clearly stated the novelty of our manuscript.
- Methods for obtaining plant extracts: we added more information to this section, including ultrasound-assisted extraction.
- Green synthesis methods of AuNPs and AgNPs from plant extracts: we added more references to this section which is almost 3 pages long.
- AuNPs and AgNPs applications in agriculture: we agree this section lacked of the need amount of references. Therefore, this section is now almost 5 pages long and contains about 84 references from the literature. Here we talk about the potential antimicrobial application of these NPs, and how that activity can be used for water treatment and pest control. In addition to that, this section briefly reviews how these NPs can work as delivery systems for nanofertilizers, nanopesticides, and nanoherbicides.
- NPs interaction with plants: here we develop harmfull effects caused by AuNPs and AgNPs due to their accumulation in plants’ tissues, as well as other issues such as NPs uptake, translocation, and trophic transfer.
- AuNPs and AgNPs in soil: similarly to the previous section, here we describe the mechanism of interaction between the NPs and soil’s surface, NPs alteration of soil microbial biomass and its impact on soil quality and sustainability. Additionally, we mention some research regarding harmful effects to other organisms such as earthworms.

Reviewer 3 Report
This review is titled “Green synthesis of metal nanoparticles from plant extracts, and their possible application as antimicrobial agents in the agricultural area”.
However, the review covers only silver and gold nanoparticles, so the title should be changed accordingly. Beside this, it has an intrinsic weakness: searching WebOfScience for “green synthesis of silver nanoparticles” one obtains 8233 hits, among which 538 are reviews, and when searching “green synthesis of gold nanoparticles” one obtains 5372 hits among which 462 are reviews.
So another review about green synthesis of silver or gold nanoparticles should be focused on something extremely new and specific, to be significant to the reader.
I agree that the “possible application as antimicrobial agents in the agricultural area” could be an interesting, significant and less common topic, however, in the present form this paper dedicates only 2,5 pages, that is largely unsufficient.
Moreover, Sections like 2. Importance of nanoparticles; 3. Conventional methods for the synthesis of gold and silver nanoparticles; 7. Nanoparticles characterization, that, with Introduction, occupy almost one half of the overall text, are honestly useless. As a matter of fact, they use a very simplified approach while dealing with concepts that have been described literally in hundreds of reviews, not to mention articles.
The paper in the present form must be rejected.
However, I suggest resubmission after complete rewriting and strengthening and deepening of potentially much more interesting sections like 5. Methods for obtaining plant extracts; 8. Gold and Silver nanoparticles applications in agroindustry; 9. NPs interactions with plants (in particular regarding accumulation in soils).
Moreover, in the introduction it must be clearly stated what makes the review different and interesting with respect to the hundreds of already published ones on the green synthesis of Ag or Au nanoparticles
Author Response
Reviewer 3: Dear reviewer, thank you for your observations. The manuscript was reviewed and restructured taking into consideration your comments. Additionally, I would like to mention you that we deleted several sections from the original manuscript as suggested by you, and we strengthen other sections such as NPs accumulation in soils, and of course, NPs applications in agricultural area, specially their antimicrobial activity. In the new manuscript you’ll find the following structure:
- Introduction: We clearly stated the novelty of our manuscript.
- Methods for obtaining plant extracts: we added more information to this section, including ultrasound-assisted extraction.
- Green synthesis methods of AuNPs and AgNPs from plant extracts: we added more references to this section which is almost 3 pages long.
- AuNPs and AgNPs applications in agriculture: we agree this section lacked of the need amount of references. Therefore, this section is now almost 5 pages long and contains about 84 references from the literature. Here we talk about the potential antimicrobial application of these NPs, and how that activity can be used for water treatment and pest control. In addition to that, this section briefly reviews how these NPs can work as delivery systems for nanofertilizers, nanopesticides, and nanoherbicides.
- NPs interaction with plants: here we develop harmfull effects caused by AuNPs and AgNPs due to their accumulation in plants’ tissues, as well as other issues such as NPs uptake, translocation, and trophic transfer.
- AuNPs and AgNPs in soil: similarly to the previous section, here we describe the mechanism of interaction between the NPs and soil’s surface, NPs alteration of soil microbial biomass and its impact on soil quality and sustainability. Additionally, we mention some research regarding harmful effects to other organisms such as earthworms.
- Conclusion.
